# Effect of Graphene Oxide Addition on the Anticorrosion Properties of the Phosphate Coatings in Neutral and Acidic Aqueous Media

**DOI:** 10.3390/ma15196588

**Published:** 2022-09-22

**Authors:** Vasily A. Bautin, Ilya V. Bardin, Askar R. Kvaratskheliya, Sergey V. Yashchuk, Evangelos V. Hristoforou

**Affiliations:** 1MISiS, National University of Science and Technology, 119049 Moscow, Russia; 2State Research Center of the Russian Federation, Central Mechanical Engineering Research Institute, Research and Production Association, Joint-Stock Company (CNIITMASH JSC), 115088 Moscow, Russia; 3PAO Severstal, 127299 Moscow, Russia; 4Laboratory of Electronic Sensors, School of Electrical and Computer Engineering, NTUA, National Technical University of Athens, 15780 Athens, Greece

**Keywords:** low carbon steel, phosphate coating, graphene oxide, corrosion, depolarization

## Abstract

**Highlights:**

**What are the main findings?**
GO improves the anticorrosion properties of the zinc phosphate coatings in neutral media, and disimproves them in acidified media.

**What is the implication of the main finding?**
The acidity of the media should be taken into account when using GO-containing zinc-phosphate coatings.

**Abstract:**

Graphene oxide (GO) is an advanced additive improving the properties of various types of coatings and intensifying the deposition process. In this work, GO is used as an additive to the traditional phosphating solution of the widely used Russian low-carbon steel 08YU (DC04). The anticorrosion properties of the obtained phosphate coatings were investigated in neutral (0.5 M NaCl) and acidified (0.1 M Na_2_SO_4_ + 0.02 M H_2_SO_4_) aqueous solutions. Increasing the GO concentration in the phosphating solution to 0.3 g/L was found to improve the anticorrosion properties of the phosphate coatings in neutral NaCl solutions. At the same time, in acidified Na_2_SO_4_ solutions, the corrosion rate of 08YU steel with phosphate coatings increased as a function of the GO concentration. It is assumed that a possible reason for various corrosive behavior is the influence of the GO plates distributed in the coating on the rate of the oxygen or hydrogen reduction reactions.

## 1. Introduction

Graphene oxide (GO) shows promise as an eco-friendly additive in different types of coating to improve their protective properties against corrosion [1,2]. By using the incorporation of GO to organic [3,4,5,6], metallic [7,8,9,10] and inorganic non-metallic coatings [11], researchers have created composite coatings with higher levels of anticorrosion properties. In this work, we have focused on well-known inorganic non-metallic coating, i.e., phosphate coating on low carbon steel. Phosphating is a traditional surface treatment for steel and galvanized frameworks [12,13]. Unfortunately, the phosphate coating alone is not able to provide full protection of the steel against atmospheric corrosion. Sealing or painting of the phosphate coating is necessary to obtain the required level of anticorrosion protection [12,13]. At present, the search for new and effective accelerators of the phosphating process, in order to decrease the process time and solution temperature, is a challenge [14,15,16,17]. Xie et al. [11] showed that GO addition accelerates the phosphating process and improves the anticorrosion properties of phosphate coating. It was suggested [11] that at the initial stages of the phosphating process GO plates located on the metallic surface are favorable places for nucleation of phosphate crystals. If the GO concentration is higher than the optimal value, anodic reaction of the iron dissolution is restricted and coating growth rate is decreased. However, it was found [11] that at any investigated GO concentration, the modified coating has a higher level of anticorrosion properties comparing to the phosphate coating obtained without GO addition.

In [11], Q235 steel with composition (wt.%) ≤0.22 C, ≤0.35 Si, ≤0.05 S, ≤0.05 P, ≤1.40 Mn, was used for investigation. Electrochemical tests were carried out only in 3.5% NaCl water solution imitated marine environment. In this paper, GO modified phosphate coatings were obtained on the surface of samples made of widely used Russian 08YU (DC04) low carbon steel, with nominal composition (wt.%): ≤0.07 C, ≤0.03 Si, ≤0.025 S, ≤0.020 P, ≤0.35 Mn. Therefore, 08YU steel has a lower concentration of C and Mn compared to Q235 steel. Electrochemical measurements of 08YU steel samples with GO modified phosphate coatings were conducted, both in 0.5 M NaCl solution and in 0.1 M Na_2_SO_4_ + 0.02 M H_2_SO_4_ solutions. The solution containing NaCl was chosen to compare results with [11]. The corrosion behavior of GO modified phosphate coatings in acidified solution was studied since some steel constructions may suffer in acidic environments, such as acid rain, mine water and other. It should be noted that the mechanism of low carbon steel corrosion in NaCl and in Na_2_SO_4_ + H_2_SO_4_ solutions is different, due to the difference of cathodic reaction, i.e., oxygen depolarization for NaCl solution and hydrogen depolarization for Na_2_SO_4_ + H_2_SO_4_ solution, respectively. Corrosion data were calculated from electrochemical measurements. It was found that increasing GO concentration in phosphating solution leads to the improvement of anticorrosion properties of phosphate coatings on 08YU steel in NaCl solutions. At the same time, the dependence of the corrosion rate on GO concentration in phosphating solution has nonmonotonic character in acid Na_2_SO_4_ + H_2_SO_4_ solutions. The possible reasons for this corrosion behavior have been discussed.

## 2. Experimental

Samples with the dimension of 50 × 25 × 0.8 mm were produced using commercially available 08YU steel sheets provided by PAO Severstal [https://www.severstal.com]. Table 1 shows the nominal chemical composition of 08YU steel according GOST 9045-93, in comparison with measured concentrations of chemical elements in the samples investigated.

The improved Hummer’s method [18] was used for preparing the GO. 1 g graphite powder 325 mesh was mixed with 120 mL concentrate H_2_SO_4_ (98%) under agitation by magnetic stirrer ECROS PE-6110 (ECROSHIM, Saint Petersburg, Russia) for 6 h. Then, the solution was allowed to stand for 16 h. A measuring flask containing the solution was immersed in an ice bath. 3 g KMnO_4_ was added into the flask and mixed by using a magnetic stirrer for 6 h. Ice was added into the ice bath during mixing. After 16 h of exposure, the solution was heated to 98 °C during 2 h. Next, water was added dropwise until the solution changed color from black-green to brown. Then, the solution was diluted by 600 mL of water and was allowed to stand for 48 h. 10 mL H_2_O_2_ (50%) was added until the solution changed color from brown to khaki. The mixture was then filtered and washed by 1M HCl solution two times, followed by ultrasonication for 90 min and centrifugation at 16,000 rpm for 30 min. Finally, the solution was washed with water four times, followed by ultrasonication for 90 min and centrifugation at 16,000 rpm for 30 min.

GO characterization was performed by confocal Raman microspectrometer Renishaw inVia Reflex (Renishaw, Gloucester, UK) using a 532 nm Nd:YAG laser (diode type excitation). Figure 1 shows the Raman spectrum recorded for the GO after the preparation based on Hummers’ method. Two strong peaks were observed at 1344 cm^−1^ and 1602 cm^−1^, which are known to be related to the D band and G band of GO [19], respectively. This confirms that the carbon sheets consist of GO. The observed GO Raman spectrum shows the prominent G peak corresponding to the first-order scattering of the E_2_g mode. The G band is broadened and shifted to 1602 cm^−1^, whereas the D band at 1344 cm^−1^ is less prominent, indicating an increase in size of the in-plane sp2 domains, possibly due to the slight reduction of GO. The D-peak at 1344 cm^−1^ is typically considered as an indicator for defects and chemical functionalization [20,21]. The surface morphology and concentrations of the chemical elements in the GO samples were obtained by Tescan Vega SBH3 (Tescan, Brno, Czech Republic) Scanning Electron Microscope (SEM) equipped with an Oxford Instruments AZtecEnergy energy-dispersive X-ray spectroscopy (EDS) system. The SEM micrographs of the GO produced are shown in inset in Figure 1. The concentrations of the chemical elements in the samples of GO obtained via EDS are shown in Table 2.

The surface of 08YU steel samples was grounded using 600 grit SiC abrasive paper. The degreasing of samples was conducted in 10.0 wt.% NaOH at 40 °C for 5 min. After washing in cold, distilled water, the samples were immersed in phosphating solution at 40 °C for 20 min, with the composition shown in Table 3. After that, the samples were washed by cold, distilled water, and dried by blowing air at room temperature. The surface microstructure of the phosphate coating was investigated by SEM.

The electrochemical behavior of the samples was investigated using the IPC Pro MF potentiostat (Volta, Saint Petersburg, Russia) with FRA2 module in 0.5 M NaCl and 0.1 M Na_2_SO_4_ + 0.02 M H_2_SO_4_ solutions at (25 ± 2) °C. The samples’ working area was 1 cm^2^. A three-electrode system was employed in which the samples served as a working electrode. Platinum and Ag/AgCl electrodes were used as the counter and reference electrodes, respectively. Potentiodynamic polarization curves were measured from the cathodic (−1000 mV) to the anodic (−100 mV) region with a scan rate of 0.3 mV/s. The corrosion current density and corrosion potential were determined from Tafel fitting. The corrosion rate was calculated according to ASTM G 102-89. The electrochemical impedance spectroscopy (EIS) was carried out at open circuit potential (OCP) in the frequency range from 0.1 Hz to 100 kHz, with the potential amplitude of 10 mV. The time waiting for the stable OCP was set to be 600 s prior to EIS measurement in 0.5 M NaCl solution, and 300 s in 0.1 M Na_2_SO_4_ + 0.02 M H_2_SO_4_ solution. The Circuits Solver software developed by Cronas (Version 3.3, 2020) was used to analyze the EIS results.

## 3. Results and Discussions

The SEM micrographs of the phosphate coatings produced in solutions with different GO concentration are shown in Figure 2. It can be seen that the size of the phosphate crystals, and crystals growth rate, depend on the GO concentration. The size of the phosphate crystals decreases with the increasing concentration of GO.

As Figure 2a,b show, the metallic surface was not fully covered by phosphate crystals after 20 min of phosphating process in the solution without GO. However, according to Figure 2c–h, the addition of GO to the phosphating solution leads to the formation of coating on all surface of the steel after 20 min of phosphating procedure. The phosphate coatings obtained in the solution with concentrations of 0.3 and 1.2 g/L GO have a similar surface morphology (Figure 2c,d,g,h). On the other hand, as Figure 2e,f show, the coating produced in the solution with a concentration 0.6 g/L GO has some specific regions where relatively large and long crystals are absent.

The cross-sections of the coatings are shown in Figure 3. Evidently, the coatings are nonuniform. Their thickness ranges for different GO concentrations (g/L) are: 0 (from 5.37 to 12.74 µm, but with bare surface areas); 0.3 (6.4–12.22 μm); 0.6 (5.22–9.45 μm) and 1.2 (5.37–5.75 μm)

The phase composition of the phosphate coatings was investigated on Rigaku Ultima IV (Rigaku, Tokyo, Japan)using CuKα (λ = 1.5406 Å) radiation in a grazing beam, the incident beam angle being 5°.

The XRD patterns of coatings obtained with and without GO in the phosphating bath are shown in Figure 4. Table 4 shows the quantitative composition of the phases in the coatings. It can be seen that the phosphate coatings without the addition of GO consist of Zn_3_(PO_4_)_2_·4H_2_O (hopeite, JCPD file No. 37-0465) and 24.6% Zn_2_Fe(PO_4_)_2_·4H_2_O (phosphophyllite, JCPD file # 29-1427). In the studied samples of coatings with GO additives from 0.3 to 1.2 g/L GO, the peak intensities of Zn_2_Fe (PO_4_)_2_·4H_2_O (phosphophyllite, JCPD file No. 29-1427) are either weak or practically not observed. The data obtained show that the GO addition changes the phase composition of the phosphate coating and affects the preferred orientation of phosphate crystal growth.

The mechanism of traditional phosphating process without GO has been widely investigated [22,23]. Firstly, the anodic dissolution of iron and cathodic evolution of hydrogen occur according net reaction:Fe + 2H^+^ → Fe^2+^ + H_2_↑ (1)

Cathodic hydrogen depolarization leads to increasing the pH and generation of PO_4_^3−^ close to metal surface:H_3_PO_4_ → H_2_PO_4_^−^ + H^+^ → HPO_4_^2−^ + 2H^+^ → PO_4_^3−^ + 3H^+^
(2)

The absence of the Zn_2_Fe (PO_4_)_2_∙4H_2_O phase in the coatings produced in the phospating solution with the GO addition is clearly described in [22,23]. The GO shields the metal surface, which leads to a decrease in the anodic dissolution of iron and the formation of Zn_2_Fe (PO_4_)_2_∙4H_2_O phase according to the reaction:2Zn^2+^ + Fe^2+^ + 2PO_4_^3−^ + 4H_2_O → Zn_2_Fe(PO_4_)_2_·4H_2_O↓ (3)

In this case, coating is formed without participation of iron ions:3Zn^2+^ + 2PO_4_^3−^ + 4H_2_O → Zn_3_(PO_4_)_2_·4H_2_O↓ (4)

Figure 5 shows potentiodynamic polarization curves of 08YU steel with phosphate coatings produced in the phosphating solution with different GO concentration. The corresponding corrosion parameters obtained from electrochemical data are given in Table 5 and Table 6. As Table 5 shows, in 0.5 M NaCl near neutral solutions, the corrosion rate decreases by about 2 times under the addition of 0.3–0.6 g/L GO to the phosphating solution. The obtained values of the corrosion rate correlate well with the data of the authors [11].

At the same time, at a GO concentration of 0.6 g/L and more, the corrosion rate in this media increases. Furthermore, as Figure 2a,b show, on the surface of the 08YU steel treated in solution without GO, there are some uncovered regions. As a result, the steel corrosion rate in this case is high. On the other hand, the corrosion rate of 08YU steel treated in solution with GO decreases because phosphate crystals and GO plates create a barrier to the penetration of oxygen dissolved in water to the steel surface to participate in the cathodic reaction:O_2_ + 2H_2_O + 4e^−^ = 4OH^−^

Another situation was observed in the corrosive solution containing sulfuric acid (see Table 6 and Figure 5b). Actually, as Figure 5b shows, the corrosion rate of the 08YU steel with phosphate coating obtained in solutions with different GO concentration has the opposite behavior.

According to Table 6, the dependence of the corrosion rate on the GO concentration for the acidified solution studied is significantly different. Figure 5b shows that corrosion of steel 08YU in acidic media occurs with hydrogen depolarization according to the reaction:2H^+^ + 2e^−^ = H_2_

In this case, there are no diffusion restrictions on the hydrogen ions movement. The hydrogen ion easily penetrates to the metal surface to participate in the cathodic reaction. However, the GO plates, in the case of their contact with a metal surface and with each other, can act as additional effective cathodes [24,25] and accelerate the corrosion rate. This is confirmed by a 3.34 times increase in the corrosion rate with the addition of 0.3 g/L GO. With a further increase in the GO content from 0.3 to 1.2 g/L, the corrosion rate increases by more than 1.5 times (see Table 6).

The EIS measurement is adopted to further characterize the electrochemical corrosion behavior of phosphate coating. All plots shown in Figure 6a and Figure 7a have a similar shape. It is a sector of the circle (curve on the graph) for neutral (Figure 6a) and semicircle for acidic solutions (Figure 7a), respectively, but with different diameters. To simulate the experimentally obtained impedance curves, two equivalent circuits were used, which correspond to the behavior of steel with a protective coating in a neutral solution (Figure 6d) and an acidic solution (Figure 7d) [26,27]. The selection of values for the elements of equivalent circuits was carried out using the ‘EIS Spectrum Analyzer Software’ [28]. Equivalent circuits are presented in the tabs of Figure 6a and Figure 7a. The diagrams and nominal values of the components in the simulation accurately describe the hodograph curves and Nequist diagrams. The calculation error does not exceed 10%. The values of the elements of the scheme for coated corrosion for a neutral medium are presented in Table 7, for an acidic one they are given in Table 8.

According to the authors of [11], the flattened capacitive sector of a circle and a semicircle indicates frequency dispersion due to phosphate defects covering [17,29]. However, as our EIS data show, this statement is valid only for the corrosion process in neutral media. As GO is added into the phosphating bath, the diameter of the circle sector increases until getting to the largest size for GO 0.3 g/L, after which it decreases as the content of GO further increases. It is known that the larger the semicircle’s size, the higher the polarization resistance [30]. This is in accordance with Bode impedance plots, as shown in Figure 6b. The impedance value |Z| exhibits the same changing tendency compared to Nyquist plots. |Z| increases as GO is added, indicating an increase in the corrosion resistance of phosphate coating in a neutral solution.

The phase angle (−θ) at high frequencies is reported to be useful to evaluate coating integrity as exposed to corrosive electrolytes [31,32,33]. It is known that the phase angle at 10 kHz indicates the change in coating integrity during exposure to a corrosive electrolyte. For an intact coating without defects, the phase angle is about –90°, whereas for steel without a protective coating, the phase angle tends to zero [31,34]. As shown in Figure 6c, a higher phase angle in the high frequency range up to 10 kHz, as well as the best corrosion resistance characteristics, were obtained on a sample with an addition of 0.3 g/L GO. The results obtained in [11] also confirm that for the samples with the GO addition, as compared to those without GO, the phosphate coating improves and the phase angle increases. In fact, the coating is denser with fewer defects.

The equivalent circuit (Figure 6a) is a modified Randall circuit containing R_0_, which is the uncompensated resistance of the electrolyte solution, and R_t_, which is the polarization resistance. Here, instead of an ideal double layer capacitor (C_dl_), a constant phase element (CPE_dl_) was used to accurately simulate the frequency distribution characteristic [35,36], as well as a W—Warburg element, which corresponds to the diffusion resistance. CPE has the impedance dispersion ratio $Z=\frac{1}{\Upsilon_{0}{\left(f\omega \right)}^{n}}$, where $\Upsilon_{0}$ and *n* are the conductivity and the empirical CPE index, respectively, *j* is the imaginary number, and ω is the angular frequency [37,38]. The resulting hodograph curves are composed of a capacitive loop at high frequency, parallel to the capacitance of the double layer, followed by a gentle tail at low frequency. The latter is assumed to represent the smoothed tail of the diffusion process described by element W, as established by the authors [27].

As can be seen from Table 7, the R_t_ values increase with decreasing CPE in the samples with the addition of GO. In this case, the maximum R_t_ value of 2983.7 Ohm·cm^2^ corresponds to the sample with the addition of 0.3 g/L GO to the phosphating solution, which shows the minimum CR value. According to the authors of [11], this is due to the fact that the corrosion behavior of steel with a phosphate coating is a diffusion control process and the addition of GO does not change the mechanism of protection against corrosion of the phosphate coating. Thus, the higher R_t_ values of samples with GO indicate better corrosion resistance than the sample without the GO addition.

However, according to the semicircle EIS data in a solution of 0.1 M Na_2_SO_4_ + 0.02 M H_2_SO_4_ on samples with the addition of GO to the phosphating bath, the diameter of the semicircle decreases with an increase in GO content (Figure 7a). This indicates not only the possibility of a decrease in the number of defects and, as a consequence, an increase in corrosion resistance in neutral media, but on the effect of GO additives on the course of electrochemical stages of the corrosion process in acid solutions.

Moreover, the shape of the hodograph curves obtained in an acidic medium has a significant difference in comparison with the curves in a neutral solution of 0.5 M NaCl. According to the authors [39,40] who have published several papers on the corrosion of pure iron exposed to H_2_SO_4_/Na_2_SO_4_ solutions with dissolved H_2_S, this bend of the hodograph at low frequencies corresponds to an inductive loop. These authors also measured impedance diagrams at different anodic potentials to facilitate anodic response versus cathodic contribution [39,40].

At higher anode potentials, an induction loop appeared at a low frequency. This prompted the authors [39,40] to use a different equivalent circuit that claims to represent anodic dissolution. In our case, we also used this scheme, which is presented on the tab of Figure 7a, since it provided the best match of the simulation results in the program EIS Spectrum Analyzer. Here, *R_0_*, *R_t_*—is the charge transfer resistance caused by the presence of a protective coating; *R_e_* is the resistance of the electrolyte; the *CPE_dl_* element corresponds to the capacitance of the electric double layer, which occurs on the rough surface of the steel in the electrolyte solution; *L* is the inductance element.

Analysis of the cathodic and anodic polarization curves (Figure 5b) clearly shows that the addition of GO to the phosphating solution affects the anodic and cathodic reactions. Indeed, GO increases the anodic dissolution of the steel and also accelerates the cathodic hydrogen evolution reaction.

Furthermore, in Figure 7a we see that the diameter of the semicircle decreases with increasing GO content. The smaller the size of the semicircle, the lower the polarization resistance [30]. This corresponds in part to the Bode impedance plot shown in Figure 7b. Impedance value |Z|, when compared with the Nyquist plots, demonstrates a non-monotonic trend for samples without and with the GO addition in the low frequency range from 10^−2^ to 10^2^ Hz. |Z| decreases as GO is added. Similar dependences of the Nyquist curves on mild steel immersed in acid media with the addition of the corrosion inhibitors were obtained by the authors [41,42]. These authors associate the increase in inhibition with a drop in the local dielectric constant and/or an increase in thickness C_dl_. In our case, by analogy, the addition of GO leads to an increase in the dielectric constant, which, as mentioned above, probably facilitates the cathodic process of electrochemical corrosion in 0.1 M Na_2_SO_4_ + 0.02 M H_2_SO_4_.

As can be seen from Table 8, the Rt values decrease with increasing GO concentration, with the CPE having the lowest value for the sample without the GO addition. This regression can be the result of an increase in the local dielectric constant and/or a decrease in the thickness of the electric double layer, which probably leads to the disinhibition of the cathodic process. The results obtained cast doubt on the earlier conclusions [11] that GO additives do not affect the corrosion mechanism and that only diffusion control is realized.

## 4. Conclusions

In this work, we investigated the use of GO obtained by the modified Hummer’s method as an additive in the process of phosphating of steel to improve its corrosion properties. The results show that GO sheets can assist the phosphating process and improve the morphology of the phosphate coating, providing better corrosion resistance. It has been found that GO sheets can be absorbed on the substrate surface and then act as sediment layers at the initial stage of the phosphating process. This can inhibit the release of metal ions and thus contribute to a change in the phase composition of the phosphate crystals.

It was found that the addition of GO to the phosphating solution makes it possible to improve the anticorrosion properties of phosphate coatings on 08YU steel only in a neutral environment. The optimal GO concentration for this case was found to be 0.3 g/L. On the other hand, in acidified media the GO addition may accelerate corrosion associated with the release of the cathodic electrochemical process; therefore, GO additives in the latter case are not recommended. The possible reason for different corrosion behavior of obtained samples in near neutral and acidified media is the effect of GO plates distributed in the coating on the rate of cathodic reaction of oxygen or hydrogen depolarization. The mechanism of the effect of GO on the rate of electrochemical corrosion is worth being further investigated and will be discussed elsewhere.

## Figures and Tables

**Figure 1 materials-15-06588-f001:**
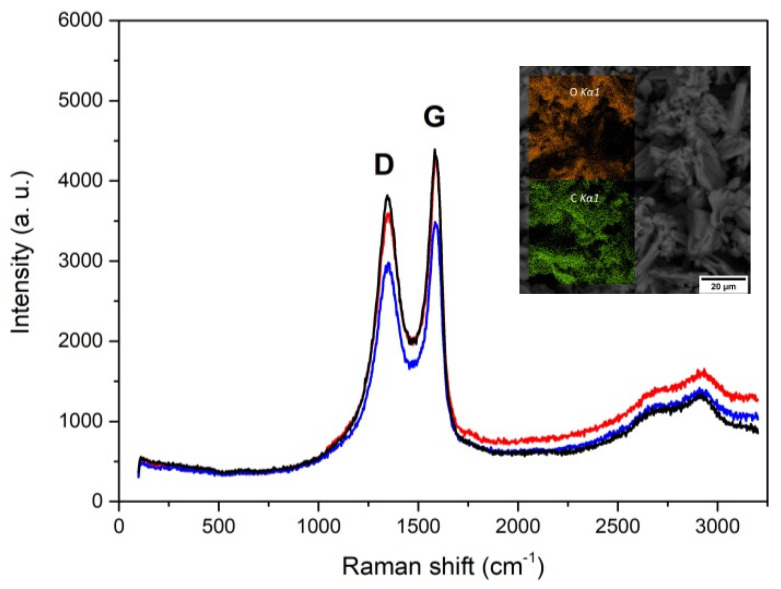
Raman spectrum of as-made GO and SEM images of GO sample with the EDS maps showing the distribution of C and O. Lines correspond to different control points.

**Figure 2 materials-15-06588-f002:**
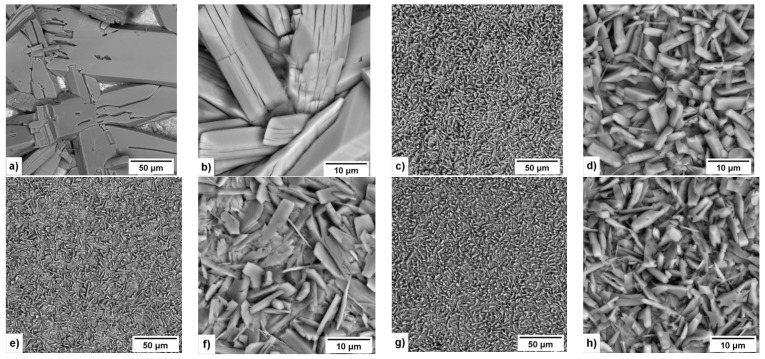
SEM images of 08YU steel with phosphate coatings obtained in solution with different concentration of GO (g/L): 0 (**a**,**b**); 0.3 (**c**,**d**); 0.6 (**e**,**f**) and 1.2 (**g**,**h**).

**Figure 3 materials-15-06588-f003:**
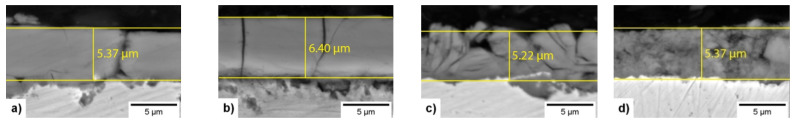
SEM images of the cross-section structure of phosphate coatings obtained in solutions with different GO concentration in the phosphating solution (g/L): 0 (**a**); 0.3 (**b**); 0.6 (**c**) and 1.2 (**d**).

**Figure 4 materials-15-06588-f004:**
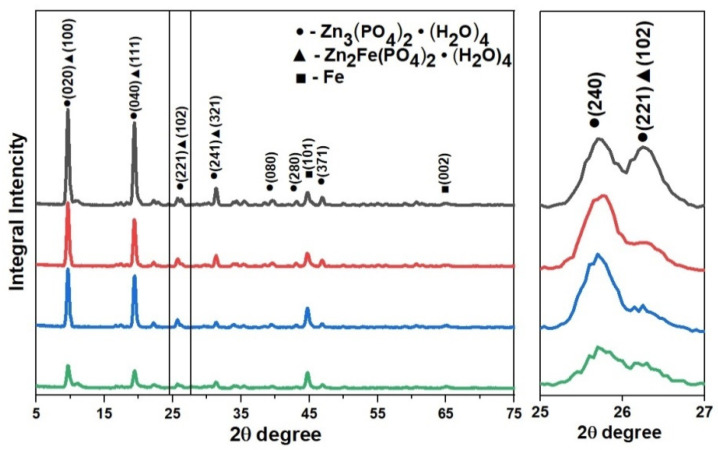
XRD patterns of phosphate coatings obtained from phosphating bath with different content of GO (top down): without adding GO to the phosphating solution (black line); with the addition of 0.3 g/L GO (red line); with the addition of 0.6 g/L GO (blue line); and with the addition of 1.2 g/L GO (green line).

**Figure 5 materials-15-06588-f005:**
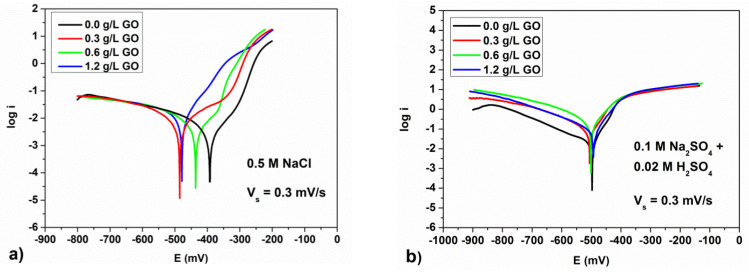
Potentiodynamic polarization curves of 08YU steel with phosphate coating produced in phosphating solution with different GO concentration for 0.5 M NaCl (**a**) and 0.1 M Na_2_SO_4_ + 0.02 M H_2_SO_4_ solution (**b**), respectively.

**Figure 6 materials-15-06588-f006:**
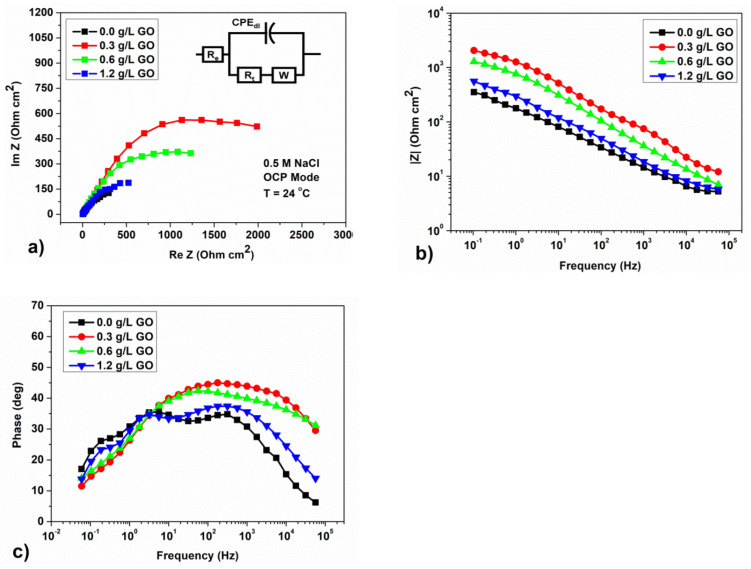
Results of measurements of electrochemical impedance for samples with different concentrations of GO in a solution of 0.5 M NaCl at OCP; (**a**) Nyquist curves and equivalent scheme for corrosion of steel with a protective coating in a solution of 0.5 M NaCl; (**b**) Bode diagrams; and (**c**) frequency dependence of the phase angle of the electrochemical impedance.

**Figure 7 materials-15-06588-f007:**
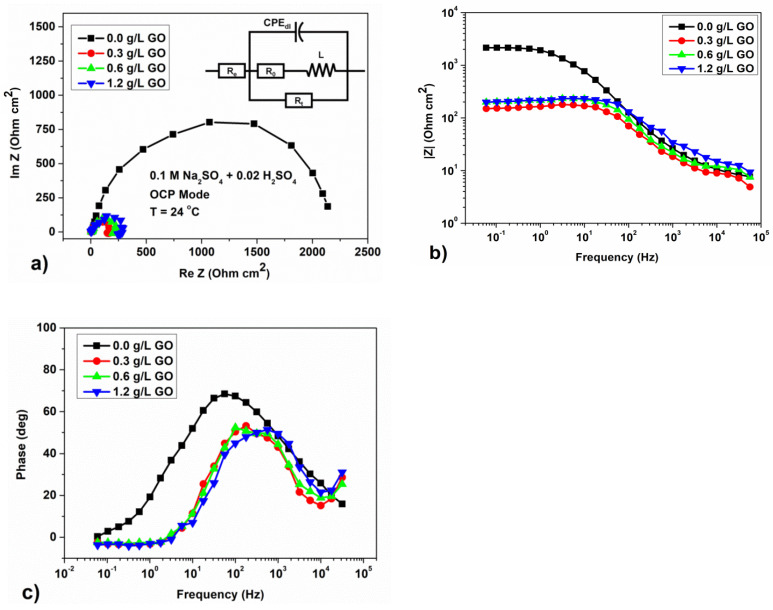
Results of measurements of electrochemical impedance for samples with different concentrations of GO in a solution of 0.1 M Na_2_SO_4_ + 0.02 M H_2_SO_4_ at OCP; (**a**) Nyquist curves and equivalent scheme for corrosion of steel with a protective coating in a solution of 0.1 M Na_2_SO_4_ + 0.02 M H_2_SO_4_; (**b**) Bode diagrams; and (**c**) frequency dependence of the phase angle of the electrochemical impedance.

**Table 1 materials-15-06588-t001:** Nominal and measured chemical composition (wt.%) of 08YU steel samples.

Composition	Fe	C	Si	S	P	Mn	Al
nominal	base	≤0.070	≤0.030	≤0.025	≤0.0200	≤0.350	0.020–0.070
measured	base	0.070	0.012	0.012	0.0096	0.298	0.025

**Table 2 materials-15-06588-t002:** Measured chemical composition (%) of GO sample.

Composition	C	O	Si	P	S	Cl
Measurement #1	74.46	25.24	0.01	0.00	0.28	0.01
Measurement #2	73.14	26.47	0.04	0.01	0.32	0.02

**Table 3 materials-15-06588-t003:** Chemical composition of phosphating solution.

Component	Zn(H_2_PO_4_)_2_	Zn(NO_3_)_2_	H_3_PO_4_	GO
Concentration, g/L	60	60	8	0; 0.3; 0.6; 1.2

**Table 4 materials-15-06588-t004:** Phase composition of the phosphate coatings.

Concentration of GO in the Phosphating Bath	Zn_3_(PO4)_2_·4(H_2_O), wt.%	Zn_2_Fe(PO_4_)_2_·4(H_2_O), wt.%
0	75.4	24.6
0.3; 0.6; 1.2	100	-

**Table 5 materials-15-06588-t005:** Corrosion parameters determined from polarization curves in 0.5 M NaCl solution.

Concentration GO, g/L	Corrosion Potential, V (Ag/AgCl)	Corrosion Current Density, mA/cm^2^	Corrosion Rate, mm/year	Corrosion Rate, mm/min
0	−0.383	7.85 × 10^−3^	0.105	1.99 × 10^−7^
0.3	−0.49	3.89 × 10^−3^	0.048	9.22 × 10^−8^
0.6	−0.427	4.44 × 10^−3^	0.053	1.01 × 10^−7^
1.2	−0.476	1.12 × 10^−2^	0.134	2.54 × 10^−7^

**Table 6 materials-15-06588-t006:** Corrosion parameters determined from polarization curves in 0.1 M Na_2_SO_4_ + 0.02 M H_2_SO_4_ solution.

Concentration GO, g/L	Corrosion Potential, V (Ag/AgCl)	Corrosion Current Density, mA/cm^2^	Corrosion Rate, mm/year	Corrosion Rate, mm/min
0	−0.493	0.0201	0.233	4.43 × 10^−7^
0.3	−0.505	0.067	0.778	1.48 × 10^−6^
0.6	−0.499	0.090	1.046	1.99 × 10^−6^
1.2	−0.482	0.100	1.162	2.21 × 10^−6^

**Table 7 materials-15-06588-t007:** The values of the elements for coated corrosion scheme for a neutral solution.

Element	Concentration GO, g/L			
	0.0	0.3	0.6	1.2
E_OCP_ (Ag/AgCl), mV	−506	−413	−411	−397
R_e_, Ω·cm^2^	3.8	4.0	3.1	3.0
R_t_, Ω·cm^2^	510.7	2983.7	1756.1	711.1
CPE_dl_, $\Upsilon_{0}$, (Ω^−1^·cm^−2^·s^n^)	1.65 × 10^−3^	2.2 × 10^−4^	3.4 × 10^−4^	8.5 × 10^−4^
n	0.46	0.48	0.5	0.45
W, Ω·s^−0.5^	69.8	60.0	60.1	68.8

**Table 8 materials-15-06588-t008:** The values of the elements for coated corrosion scheme for an acid solution.

Element	Concentration GO, g/L			
	0.0	0.3	0.6	1.2
E_OCP_ (Ag/AgCl), mV	−551	−506	−502	−510
R_e_, Ω·cm^2^	7.67	7.15	7	7
R_t_, Ω·cm^2^	4361.8	219.7	320.5	399.9
CPE_dl_, $\Upsilon_{0}$, (Ω^−1^·cm^−2^·s^n^)	4.75 × 10^−5^	1.02 × 10^−4^	1.07 × 10^−4^	1 × 10^−4^
n	0.78	0.74	0.71	0.68
R_0_, Ω·cm^2^	4891	583	572	599.6
L, H	9.8 × 10^−6^	25	25	24.56

## Data Availability

The data used to support the findings of this study are available from the corresponding author upon request.

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
