# Peer review of "Effect of Graphene Oxide Addition on the Anticorrosion Properties of the Phosphate Coatings in Neutral and Acidic Aqueous Media"

_materials, 2022, doi:10.3390/ma15196588_

Round 1
Reviewer 1 Report
In this manuscript, the authors reported the anticorrosive properties of the phosphate coatings were investigated in neutral and acidified aqueous solutions. It was found that an increase in GO concentration in the phosphating solution improves the anticorrosive properties of phosphate coatings in neutral NaCl solutions. The corrosion rate increases in acidified solutions. It is assumed that a possible reason for this corrosive behavior is the influence of the GO plates distributed in the coating on the rate of the cathodic oxygen or hydrogen depolarization reaction.
However, there are many inaccuracies and defects in the manuscript. Additional analysis and explanation are needed to clarify the results and discussion to further improve this paper. I don't recommend the paper to be published unless the authors show sufficient evidence to support their novelty and the significance in this study. Detailed comments are shown in the following.
1. This manuscript is in poor written, including the language and the framework in this paper. The authors are recommended to check the manuscript carefully to avoid some inaccuracies.
2. What is the novelty of this paper? While the manuscript is in general sound, the novelty of the manuscript is unclear.
3. Why were the phosphate coatings obtained in solutions with the concentrations of 0.6 g/L and 1.2 g/L in the phosphating solution thinner than that in 0.3 g/L (in Fig. 3)?
4. The corrosion resistance was measured in this manuscript with the electrochemical methods. However, more sufficient evidence should be added. For example, in reference Steel Research International, 2022, 93(4): 2100415, the morphology of the surface were observed in the electrochemical measurements. Please revise and discuss this.
5. The Abstract and Conclusion sections also need to be further revised and summarized.

Author Response
Reviewer #1
- This manuscript is in poor written, including the language and the framework in this paper. The authors are recommended to check the manuscript carefully to avoid some inaccuracies.
We carefully rechecked the text of the manuscript and made corrections.
- What is the novelty of this paper? While the manuscript is in general sound, the novelty of the manuscript is unclear.
The main message of this work is the addition of GO does not always have a positive effect on the protective properties of the phosphate coatings as previously reported. We have shown that if in a neutral environment the effect of the additive is positive, then in an acidic one, on the contrary, it is negative. We have tried to study the differences in the mechanism of influence GO on the protective properties of the phosphate coatings in neutral and acidic aqueous media. This is the novelty of this work.
- Why were the phosphate coatings obtained in solutions with the concentrations of 0.6 g/L and 1.2 g/L in the phosphating solution thinner than that in 0.3 g/L (in Fig. 3)?
Phosphate coatings obtained in solutions with the concentrations of 0.6 g/L and 1.2 g/L GO are more uniform in thickness. If in 0 g/L GO solution coating thickness ranges from 5.37 to 12.74 µm (with bare surface areas) in 1.2 g/L GO solution - 5.37-5.75 μm. In this case, it is not worth comparing the coatings obtained in different electrolytes based on the average coating thickness. Therefore, we did not make such a comparison.
- The corrosion resistance was measured in this manuscript with the electrochemical methods. However, more sufficient evidence should be added. For example, in reference Steel Research International, 2022, 93(4): 2100415, the morphology of the surface were observed in the electrochemical measurements. Please revise and discuss this.
At present Tafel fitting and electrochemical impedance spectroscopy (EIS) are the standard laboratory practices to compare corrosion behavior of different materials and coatings. The study of the surface morphology of samples after anodic and cathodic polarization does not provide additional information, since the state of the surface after polarization differs significantly from the surface during free corrosion without the application of an external current. The morphology of the surface of the samples after the EIS without the significant polarization practically does not change due to the insignificant duration of the corrosive effect of the medium.
- The Abstract and Conclusion sections also need to be further revised and summarized.
The Abstract and Conclusion have been corrected.

Author Response
Reviewer #2
1- The highlights and title are long and need to be shortened.
The highlights and title have been changed.
2- Use word abstract instead of Annotation.
Adjustments have been made
3- Use the sentence case in the keywords.
Letter case has been changed.
4- Most cited paper are too old, not updated, it is not usual to cite paper from 2013 and leave 2019.
There are not many publications on this topic. We tried to include the most important ones. At the same time, we believe that the year of publication of an article does not determine its scientific value.
5- Line 98 E2g not E2g
6- Line 354, Hummer, not Hammer.
7- Line 366, Therefore, GO additives cannot be used. This sentence is not well understood.
Adjustments have been made
